# Effects of the Preferential Oxidation of Phenolic Lignin Using Chlorine Dioxide on Pulp Bleaching Efficiency

**DOI:** 10.3390/ijms232113310

**Published:** 2022-11-01

**Authors:** Yi Liu, Baojuan Deng, Jiarui Liang, Jiao Li, Baojie Liu, Fei Wang, Chengrong Qin, Shuangquan Yao

**Affiliations:** Guangxi Key Laboratory of Clean Pulp & Papermaking and Pollution Control, School of Light Industrial and Food Engineering, Guangxi University, Nanning 530004, China

**Keywords:** lignin, chlorine dioxide, oxidation, bleaching, pulp

## Abstract

Chlorine dioxide is widely used for pulp bleaching because of its high delignification selectivity. However, efficient and clean chlorine dioxide bleaching is limited by the complexity of the lignin structure. Herein, the oxidation reactions of phenolic (vanillyl alcohol) and non-phenolic (veratryl alcohol) lignin model species were modulated using chlorine dioxide. The effects of chlorine dioxide concentration, reaction temperature, and reaction time on the consumption rate of the model species were also investigated. The optimal consumption rate for the phenolic species was obtained at a chlorine dioxide concentration of 30 mmol·L^−1^, a reaction temperature of 40 °C, and a reaction time of 10 min, resulting in the consumption of 96.3% of vanillyl alcohol. Its consumption remained essentially unchanged compared with that of traditional chlorine dioxide oxidation. However, the consumption rate of veratryl alcohol was significantly reduced from 78.0% to 17.3%. Additionally, the production of chlorobenzene via the chlorine dioxide oxidation of veratryl alcohol was inhibited. The structural changes in lignin before and after different treatments were analyzed. The overall structure of lignin remained stable during the optimization of the chlorine dioxide oxidation treatment. The signal intensities of several phenolic units were reduced. The effects of the selective oxidation of lignin by chlorine dioxide on the pulp properties were analyzed. Pulp viscosity significantly increased owing to the preferential oxidation of phenolic lignin by chlorine dioxide. The pollution load of bleached effluent was considerably reduced at similar pulp brightness levels. This study provides a new approach to chlorine dioxide bleaching. An efficient and clean bleaching process of the pulp was developed.

## 1. Introduction

Global industrialization has led to the enhancement of prosperity and development of human society [1,2]. However, this has resulted in an increase in environmental pollution, causing significant disruptions in human daily lives as well as in industrial production [3]. Sustainable development is the natural choice for modern industrial development [4,5], and clean production is an integral component of this strategy [6,7]. A significant amount of the produced primary raw material yields from pulp and paper industries [8,9]. Therefore, the cleaner production of pulp and paper is of significant importance for sustainable development [10]. The production of these materials comprises three main stages: cooking, bleaching, and papermaking. Among these, bleaching is the central step [11]. Elemental chlorine-free (ECF) bleaching using chlorine dioxide is the primary bleaching method employed in mainstream bleaching technology [12]. This is because of the excellent selectivity of chlorine dioxide for the delignification of lignin [13,14]. However, small amounts of adsorbable organic chlorides (AOX) [15] are generated during the bleaching process, owing to the chlorination of lignin by-products occurring during the oxidation process [16]. Thus, the production yield of clean pulp and paper production is reduced. Therefore, clean and efficient chlorine dioxide bleaching methods for pulp must be developed.

Researchers have attempted to reduce the production of AOX at the source [17,18]. Hypochlorite was generated owing to the decomposition of chlorine dioxide during the bleaching process [19,20]. Thus, the oxidation and chlorination of lignin occurred simultaneously. AOX production can be effectively reduced via pulp pretreatment [21,22], high temperature chlorine dioxide bleaching [23,24], high concentration chlorine dioxide bleaching, and the addition of hypochlorite inhibitors [25,26,27]. The essence of these methods is to reduce the lignin content of the unbleached pulp and increase the rate of oxidation of lignin with chlorine dioxide. This results in the inhibition of lignin chlorination. However, the influence of the structural characteristics of lignin on the efficiency of AOX production and bleaching reactions has not been explored. The use of lignin model compounds to simulate the complex chemistry of lignin has proven to be an effective method. Ni et al. [28] investigated the reaction of four different lignin model compounds, namely, veratryl alcohol, veratrol, vanillyl alcohol, and creosol, with chlorine dioxide. The reaction rate of chlorine dioxide with phenolic lignin units was 10^5^ times higher than that with non-phenolic lignin units. Lachenal et al. [29] found that non-phenolic structural units compete with phenolic structural units in the delignification of chlorine dioxide. Shi et al. [30] compared the reactivity of the phenolic and non-phenolic lignin model compounds, vanillyl alcohol and veratryl alcohol, respectively, with chlorine dioxide. The activation energy of the reaction of quinoa with chlorine dioxide was more than 50 times higher than that of the reaction of vanillyl alcohol with chlorine dioxide. Additionally, more AOX were generated from the reaction of veratryl alcohol with chlorine dioxide [30,31]. The results of this study indicate that the reaction of phenolic and non-phenolic structural units of lignin with chlorine dioxide not only produces a difference in reaction rates but also results in the production of different AOX. Therefore, the phenolic structure of lignin is preferentially consumed through the regulation of the reaction environment. The pollution load of bleaching wastewater is reduced when efficient bleaching is achieved. In addition, this is conducive to the improvement of the physical and chemical properties of pulp fiber. This study will be of great significance to reduce pulp bleaching costs and realize cleaner production.

Herein, vanillyl alcohol and veratryl alcohol were used as phenolic and non-phenolic lignin model substances, respectively. The effects of chlorine dioxide concentration, reaction temperature, and reaction time on the consumption of vanillyl alcohol and veratryl alcohol were investigated. The physicochemical properties of the pulp before and after lignin oxidation by different chlorine dioxides were characterized using scanning electron microscopy (SEM), X-ray diffraction (XRD), X-ray photoelectron spectroscopy (XPS), and thermogravimetric analysis (TGA). The structure of lignin in different pulps was analyzed using two-dimensional heteronuclear single quantum coherence nuclear magnetic resonance (2D-HSQC NMR) measurements. Furthermore, the effects of the modulation of the reaction consumption of lignin phenolic and non-phenolic structural units on the pulp properties, amount of AOX produced, and composition were analyzed. This study presents a new method for the efficient, clean bleaching of pulp.

## 2. Results and Discussion

### 2.1. Oxidative Consumption of Vanillyl Alcohol and Veratryl Alcohol by Chlorine Dioxide

The concentration of chlorine dioxide is an important factor affecting the consumption of lignin in the reaction [32]. Therefore, variation in consumption yields of vanillyl alcohol and veratryl alcohol with respect to chlorine dioxide concentration was investigated. The reaction temperature was 60 °C, and the reaction time was 30 min; the reaction substrates were 10 mmol·L^−1^ vanillyl alcohol and 10 mmol·L^−1^ veratryl alcohol. In fact, equimolar quantities were investigated to maximize the competition between the two models with chlorine dioxide. The decomposition of chlorine dioxide is limited. The chlorine dioxide concentrations were 15, 20, 25, 30, 35, 40, and 45 mmol·L^−1^. The results are shown in Figure 1.

The consumption of vanillyl alcohol increased rapidly with increasing chlorine dioxide concentration up to 30 mmol·L^−1^. The consumption yield increased significantly from 54.4% to 91.8%. This was attributed to the higher oxidative activity of the phenolic lignin than the non-phenolic lignin [31]. The reactivity of vanillyl alcohol with chlorine dioxide remained stable at chlorine dioxide concentrations higher than 30 mmol·L^−1^. Notably, 93.2% of vanillyl alcohol was consumed at 45 mmol·L^−1^ chlorine dioxide concentration. Furthermore, the consumption of veratryl alcohol during the reaction with chlorine dioxide showed a two-phase trend with chlorine dioxide concentration. Veratryl alcohol consumption increased slowly at low concentrations, exhibiting an increase from 11.02% to 25.6% at 30 mmol·L^−1^ chlorine dioxide concentration. However, veratryl alcohol consumption increased rapidly at high chlorine dioxide concentrations. The consumption yield of veratryl alcohol increased rapidly to 73.9% at 45 mmol·L^−1^ chlorine dioxide concentration. This was because the reaction rate of veratryl alcohol with chlorine dioxide was lower than that of vanillyl alcohol with chlorine dioxide at low concentrations. Vanillyl alcohol was rapidly consumed as chlorine dioxide concentration increased. Thus, the competitive reaction with veratryl alcohol was reduced. Consequently, the reaction rate of veratryl alcohol with chlorine dioxide increased at high concentrations. Vanillyl alcohol is rapidly consumed, and veratryl alcohol consumption is suppressed at a chlorine dioxide concentration of 30 mmol·L^−1^. Therefore, the optimum chlorine dioxide concentration was estimated as 30 mmol·L^−1^.

The reaction of phenolic or non-phenolic lignin structures with chlorine dioxide is classified as an oxidation reaction. This reaction can be promoted by increasing the reaction temperature. Therefore, the effects of temperature on the oxidation of vanillyl alcohol and veratryl alcohol by chlorine dioxide were studied. The following temperatures were considered; 25, 30, 35, 40, 45, 50, 55, and 60 °C. The concentrations of chlorine dioxide, vanillyl alcohol, and veratryl alcohol were 30, 10, and 10 mmol·L^−1^, respectively, for a reaction time of 30 min. The results are shown in Figure 2. A higher rate of vanillyl alcohol consumption was obtained at low temperatures. The consumption increased from 61.3% at 25 °C to 96.8% at 40 °C. This rate of consumption then decreased with reaction temperatures above 40 °C, yielding 91.8% at 60 °C. This is attributed to the decomposition of chlorine dioxide, which is promoted at high temperatures [33]. The negative result due to the insufficient decomposition of chlorine dioxide counteracted the promotion of the reaction between chlorine dioxide and vanillyl alcohol with increasing temperature. Eventually, the rate of vanillyl alcohol consumption decreased at high temperatures. Additionally, more chlorine dioxide was used to facilitate the consumption of vanillyl alcohol at low temperatures. Consequently, the effective reaction of veratryl alcohol with chlorine dioxide occurred at a low consumption rate. The rate of consumption of veratryl alcohol slowly increased from 10.3% at 25 °C to 26.7% at 40 °C. The insufficient decomposition of chlorine dioxide also reduced veratryl alcohol consumption. The consumption rate decreased to 25.6% at 60 °C. Based on these results, the optimum reaction temperature to effectively consume the phenolic structure and inhibit the reaction of the non-phenolic structure is 40 °C.

Generally, variability in multiple reactions is obtained by modulating the reaction time. Therefore, the effect of reaction time on the rate of vanillyl alcohol and veratryl alcohol consumption was investigated. The following reaction times were investigated; 2, 4, 6, 8, 10, 15, 20, 25, and 30 min. The concentrations of chlorine dioxide, vanillyl alcohol, and veratryl alcohol used in this study were 30, 10, and 10 mmol·L^−1^, respectively. The reaction temperature was set to 40 °C. The results are shown in Figure 3. The rapid consumption of vanillyl alcohol was observed within 10 min of the commencement of the reaction. The vanillyl alcohol consumption increased from 64.0% to 96.3% after 10 min, owing to the rapid oxidation of the phenolic structural unit with chlorine dioxide [34]. The vanillyl alcohol consumption remained primarily unchanged after 10 min, owing to the large amount of vanillyl alcohol consumed in the first 10 min. This indicates that the first 10 min of the reaction is an effective depletion phase for vanillyl alcohol. The reactions between veratryl alcohol, vanillyl alcohol, and chlorine dioxide proceeded simultaneously. The consumption rate of veratryl alcohol increased with reaction time, increasing gradually from 9.9% at 2 min to 17.3% after 10 min. This is because the reaction rate of vanillyl alcohol with chlorine dioxide was better than that of veratryl alcohol with chlorine dioxide. The competitive reaction between them disappeared after vanillyl alcohol was consumed in large quantities. The consumption rate of veratryl alcohol increased with increasing reaction time after 10 min. After 30 min, it increased to 26.7%. Under optimum conditions, vanillyl alcohol should be consumed in large quantities, whereas the consumption of veratryl alcohol should be suppressed. Therefore, the optimum reaction time was determined to be 10 min.

The optimum conditions for the preferential consumption of phenolic lignin models consisted of a chlorine dioxide concentration of 30 mmol·L^−1^, a reaction temperature of 40 °C, and a reaction time of 10 min. Notably, 96.3% of the vanillyl alcohol was consumed. Its consumption remained essentially unchanged compared with that of traditional chlorine dioxide oxidation. However, the consumption of veratryl alcohol was significantly reduced from 78.0% to 17.3%, indicating the differential oxidation of phenolic and non-phenolic lignin model substances. The optimized depletion of phenolic lignin model compounds was achieved by modulating chlorine dioxide oxidation treatment.

### 2.2. Differences in Oxidation Products

In addition to the differential oxidation of the aforementioned model substances possessing different consumption rates, the composition of the oxidation products was altered. Figure 4 shows the composition of the different hydrolysates. Specific component attributes and their contents are listed in Table 1. 3,4-Dimethoxybromobenzene, veratryl alcohol, vanillyl alcohol, and 6-chloro-veratryl alcohol were detected during the traditional chlorine dioxide oxidation, and their retention times were 29.50, 36.39, 38.53, and 41.46 min, respectively. 3,4-Dimethoxybromobenzene was used as the internal standard. The changes in the contents of other components were analyzed before and after oxidation. The relative contents of vanillyl alcohol and veratryl alcohol were 0.11 and 0.56, respectively. This suggests that the depletion of phenolic and non-phenolic lignin structures is not selective during traditional chlorine dioxide oxidation treatment. However, the relative content of veratryl alcohol increased to 0.96, while the relative content of vanillyl alcohol remained constant under the optimized chlorine dioxide oxidation conditions. The non-phenolic structure was protected by the modulation of the chlorine dioxide oxidation treatment. Furthermore, 6-chloro-veratryl alcohol, the relative content of one of the main products in the hydrolysate, traditional chlorine dioxide oxidation and optimized chlorine dioxide oxidation, were 0.53 and 0.30, respectively. 6-Chloro-veratryl alcohol (chlorobenzene) is derived from the chlorination of veratryl alcohol with chlorine dioxide [35], owing to the generation of chlorine gas and hypochlorite during the chlorine dioxide oxidation of lignin. The chlorinated replacement of the lignin benzene ring occurred. This is the key to the generation of organic pollutants during the chlorine dioxide oxidation of pulp [36]. Thus, the efficient and clean oxidation of chlorine dioxide was impeded. The chlorination reaction was effectively suppressed by modulating the oxidation of phenolic and non-phenolic structures with chlorine dioxide. The results showed that in the non-phenolic lignin model, veratryl alcohol was effectively retained under the optimized chlorine dioxide oxidation conditions. Additionally, the insufficient chlorination of the non-phenolic structure was prevented. The structure of lignin is one of the main factors affecting the compact structure of wood fibers. Therefore, the effect of this novel chlorine dioxide oxidation treatment on the physicochemical properties of pulp fibers was analyzed.

The main application of chlorine dioxide is to enhance brightness while preserving the viscosity of pulps [37]. Additionally, the production of organic pollutants during this process should be considered. Therefore, the effects of different chlorine dioxide oxidation treatments on pulp performance and wastewater pollution loads were investigated. The results are presented in Table 2. Compared to the traditional chlorine dioxide oxidation treatment, the optimization of the chlorine dioxide oxidation treatment resulted in pulp with a higher Kappa number in addition to an increased viscosity at similar pulp brightness. This was attributed to the high reactivity of phenolic lignin, and more non-phenolic lignin was retained in pulp fibers. In particular, the ineffective degradation of cellulose was reduced by rapid selective delignification. Its molecular chain was protected. The viscosity of the pulp fiber is less impacted compared with the traditional chlorine dioxide oxidation. In principle, chlorine dioxide only affects the chemical component of the pulp (lignin). However, the pH of the reaction solution was adjusted from 3.0 to 4.0 to minimize the ineffective decomposition of chlorine dioxide. This results in the depolymerization of cellulose and hemicellulose in the pulp. The bleaching loss of pulp was 5.0% during the traditional chlorine dioxide bleaching. However, carbohydrate depolymerization is limited during optimizing the chlorine dioxide optimized treatment process. This was attributed to shorter reaction times. As a result, the viscosity of the pulp increases. Additionally, a decrease in AOX production is observed owing to the lower amount of non-phenolic lignin involved in the chlorine dioxide oxidation reaction. These results indicate that the production of organic chlorides was limited during the effective bleaching of the pulp owing to the optimization of chlorine dioxide oxidation.

### 2.3. Analysis of Physicochemical Properties of Pulp Fibers

The physicochemical properties of pulp fibers mainly include surface morphology, fiber crystallinity index (CrI), pyrolysis properties, and surface element distribution. Figure 5 shows the microscopic morphology of eucalyptus pulp fibers before and after different chlorine dioxide oxidation treatments. The untreated eucalyptus pulp had a more complete and gently sloping surface. This is attributed to the close packing between the fibers owing to the presence of lignin and hemicellulose. Large amounts of lignin are removed by traditional chlorine dioxide oxidation treatment. Additionally, hemicellulose is degraded under prolonged acid treatment conditions [38], with the intensification of the cracking of the fibers. Rough fiber surfaces and cellulose fractionation were observed. The bleaching loss of pulp was 5.0% during the chlorine dioxide bleaching process. Therefore, changes in fiber microstructure were hypothesized. However, the phenolic lignin was selectively removed under the optimized chlorine dioxide oxidation conditions. The retained, non-phenolic lignin is linked to carbohydrates [39]. Additionally, the acidolysis degradation of carbohydrates was inhibited at short reaction times. Consequently, intact and partially exposed fiber surfaces were observed.

The CrI values of the different eucalyptus pulps are presented in Figure 6. The CrI of the pulp was reduced from 70.7% to 69.5% after traditional chlorine dioxide oxidation. This is due to the depolymerization of cellulose, accompanied by the oxidative removal of lignin under traditional chlorine dioxide oxidation conditions. The changes in the crystalline and non-crystalline zones were offset. However, the CrI of the pulp after the optimization of chlorine dioxide oxidation was high at 73.3%. This implies that the insufficient depolymerization of cellulose was suppressed by optimizing the chlorine dioxide oxidation treatment to achieve selective oxidative degradation of the lignin structure.

The maximum weight loss temperature of the slurry changed after different chlorine dioxide treatments. It decreased from 278 to 261 °C under the traditional chlorine dioxide oxidation conditions (Figure 7a), owing to the acidic degradation of cellulose during the bulk removal of lignin [40]. Consequently, a reduction in the maximum weight loss, from 67% to 53%, was observed. The maximum weight loss temperature of the pulp after the optimization of chlorine dioxide oxidation was close to that of the untreated pulp. This indicates that the cellulose structure was protected during the selective oxidative removal of lignin with a maximum weight loss of 58%. Additionally, the pyrolysis residues of the three samples, untreated, optimized, and traditional, were 11%, 17%, and 19%, respectively. This indicated that the untreated pulp had the lowest relative lignin content. A part of the lignin structure was retained during the optimization of the chlorine dioxide oxidation process. Therefore, its pyrolytic residue content was higher than that of the pulp after traditional chlorine dioxide oxidation. Figure 7b shows that the temperature at which maximum weight loss occurred decreased after multiple chlorine dioxide oxidations. This is because the structure of the cellulose is partially destroyed due to the acidic chlorine dioxide environment. However, the maximum weight loss temperature of 333 °C for the optimized chlorine dioxide oxidized pulp was higher than the maximum weight loss temperature of 326 °C for the traditional chlorine dioxide oxidized pulp. The results indicate that the structural strength of cellulose is enhanced by the retention of part of the lignin structure after the optimized chlorine dioxide oxidation treatment.

The distribution of elements on the fiber surface was altered by the component content. Figure 8 shows that the C1, C2, C3, and C4 contents of the untreated pulp are 27.3%, 60.3%, 10.5%, and 1.9%, respectively. This indicated that the untreated pulp rich in cellulose contained a small amount of lignin. Moreover, it has an O/C ratio of 0.6. The C1 content of the traditional chlorine dioxide oxidized pulp increased to 33.5% compared to that of the untreated pulp (27.3%). Therefore, the highly efficient oxidation treatment of lignin by chlorine dioxide has been demonstrated [41]. However, the C2 content was reduced from 60.3% to 54.0%, indicating that the degradation of cellulose was insufficient. This resulted in a reduction in the O/C ratio from 0.6 to 0.5. The C1 content of the optimized chlorine dioxide oxidized pulp was reduced to 24.7% compared to that of the traditional chlorine dioxide oxidized pulp (33.5%), whereas the C2 content increased from 54.0% to 62.4%. In comparison, the untreated slurry displayed a C2 content of 60.3%. A maximum O/C ratio of 0.6 was obtained, owing to the partial removal of lignin via the optimized chlorine dioxide oxidation treatment. Next, the structural transformation of lignin in the pulp was analyzed.

### 2.4. Structural Transformation of Lignin in Pulp

Figure 9 shows the structural characteristics of lignin in different pulps. The lignin methoxyl (-OCH_3_, OMe) signal is observed in the untreated pulp at δC/δH 55.6/3.73 ppm. The β-O-4 structure of the side chain region Cγ-Hγ is located at δC/δH 59.5/3.63 ppm. The β-β structure of Cα-Hα is located at δC/δH 84.9/4.64 ppm; Cβ-Hβ is located at δC/δH 53.5/3.05 ppm, and Cγ-Hγ is located at δC/δH 71.0/3.79 and 4.16 ppm. The β-O-4 structure was not observed. This indicates that a large number of aryl ether bonds were violently degraded during the sulfate cooking process [42]. However, the β-β structure is retained as a more stable C-C bond. The benzene ring regions S_2/6_ and S’_2/6_ are located at δC/δH 104.0/6.72 ppm and 106.3/7.21 ppm, respectively. G_5_ is located at δC/δH 114.8/6.68 ppm. This is consistent with the typical structure of residual lignin in the unbleached eucalyptus pulp. After traditional chlorine dioxide oxidation treatment, the OMe signal of residual lignin is observed at δC/δH 55.6/3.73 ppm. The β-O-4, β-β, and β-5 signals disappeared. Additionally, the characteristic G- and S-unit signals in the benzene ring region are not observed. This indicates that lignin was efficiently oxidized during the traditional chlorine dioxide oxidation treatment. The signals of residual lignin after the optimization of the chlorine dioxide oxidation treatment were similar to those of untreated lignin. However, the signal intensities of the G- and S-units were significantly low. This indicates that the phenolic structures were depleted owing to the optimized chlorine dioxide oxidation treatment. Therefore, the structural transformation of lignin in pulp fiber is consistent with the reaction mechanism of the chlorine dioxide oxidation model species.

## 3. Materials and Methods

### 3.1. Materials and Reagents

Eucalyptus pulp (kappa number 19.2, brightness 29.0% ISO, see Table 2) and chlorine dioxide solution (available chlorine concentration 16.7 g·L^−1^) were obtained from a local factory (Guangxi, China). It is produced using the sodium chlorate electrolysis process. Vanillyl alcohol and veratryl alcohol were obtained from Sigma-Aldrich Chemical Co. (St. Louis, MO, USA). Other analytical chemicals were purchased from Aladdin Biochemical Technology Co., Ltd. (Shanghai, China).

### 3.2. Oxidation of Lignin by Chlorine Dioxide

The differential oxidation of vanillyl alcohol and veratryl alcohol was performed using chlorine dioxide. The method and procedure were as follows: 10 mmol·L^−1^ vanillyl alcohol and 10 mmol·L^−1^ veratryl alcohol were added to a reactor (Ready Pilot Laboratory reactor, R.B. Radley & Co., Ltd., Essex, UK). Subsequently, different concentrations of aqueous chlorine dioxide solutions were added. Vanillyl alcohol and veratryl alcohol were oxidized at different temperatures and times. The pH of the reaction solution was adjusted to 3.5 by adding a pH buffer solution. The reaction was terminated by adding an appropriate amount of anhydrous sodium sulfite solution. The reaction solutions were analyzed for the presence of vanillyl alcohol and veratryl alcohol using high-performance liquid chromatography (Agilent 1260, Agilent, Palo Alto, CA, USA). For this purpose, 10 μL of each sample was injected at a flow rate of 0.5 mL·min^−1^ into the column oven set at 35 °C, and an ultraviolet detector operating at 280 nm was employed [43]. Elution conditions were ultrapure water (with 0.1% formic acid) and acetonitrile (90/10, *v*/*v*), elution time was 20 min. The reaction consumption of vanillyl alcohol and veratryl alcohol was calculated.

The reaction consumption of vanillyl alcohol and veratryl alcohol was compared under traditional chlorine dioxide oxidation conditions [44]. The same amount of the lignin model compounds (10 mmol·L^−1^ vanillyl alcohol and 10 mmol·L^−1^ veratryl alcohol) was added to 50 mmol·L^−1^ chlorine dioxide. The reaction was performed at 70 °C for 60 min. The pH of the reaction solution was adjusted to 3.5 by adding a pH buffer solution. The contents of the residual models and their consumption yields were analyzed and calculated as described above.

### 3.3. Organic Components in the Reaction Solution

The organic components in the reaction solution were extracted using ethyl acetate. The composition of the derivatization reaction solution was analyzed using a gas chromatograph (7890B GC/5977A, Agilent Technologies, Santa Clara, CA, USA). The injection volume was 1 μL. An HP-5MS column with dimensions of 30 m × 250 μm × 0.25 μm was used. The flow rate was 1 mL·min^−1^. The injection port temperature was 250 °C. The temperature of the MSD ion source was maintained at 230 °C. The MS quadrupole temperature was set to 150 °C. The MSD transmission-line temperature was set to 280 °C. The specific operations and procedures are described in a previous paper [45].

### 3.4. Chlorine Dioxide Oxidation of Pulp

The reaction of the pulp with chlorine dioxide was performed in a reactor (Ready Pilot Laboratory reactor, R.B. Radley and Co., Ltd., Essex, UK). The ratio of the pulp to pure water was 1/10 (*w*/*v*). Pulp lignin was removed under differential and traditional chlorine dioxide oxidation conditions. The traditional chlorine dioxide oxidation was carried out on 20 g pulp (dry basis) at 10% fiber concentration using 2.0% chlorine dioxide (w ClO_2_/w pulp), at pH 3.5 and 70 °C for 60 min. The differential chlorine dioxide oxidation was carried out on 20 g pulp (dry basis), at 10% fiber concentration using 0.5% chlorine dioxide (w ClO_2_/w pulp), at pH 3.5 and 40 °C for 30 min. The pulp was thoroughly mixed at 200 rpm. The solids and liquids were then separated at the end of the reaction. The reaction solution was collected and analyzed [21]. After the reaction, the pulp was washed with deionized water until a neutral pH was achieved and air-dried for storage.

### 3.5. Structural Characteristics of Lignin in Pulp

The structure of lignin in the pulp before and after oxidation with different chlorine dioxide concentrations was analyzed using 2D-HSQC NMR (Advance III HD 500 MHz, Bruker BioSpin AG, Faellanden, Switzerland) [2,46]. In fact, lignin in pulp before and after chlorine dioxide oxidation was extracted using the milled wood lignin (MWL) method [47]. However, the difficulty of lignin extraction was increased due to the low lignin content in the pulp after chlorine dioxide oxidation treatment. Fortunately, lignin of sufficient content and structural integrity was extracted by MWL treatment of large quantities of pulp. The extraction yield of lignin was 48.4%. The structural integrity of lignin is a hypothesis.

### 3.6. Physical and Chemical Characterization of the Pulp

The surface morphology of the pulp was observed using SEM (SU8220, Hitachi, Tokyo, Japan) at an accelerating voltage of 10 kV. The fiber crystallinity of the pulp was analyzed using XRD (MiniFlex600, Rigaku, Osaka, Japan) at a scanning range of 5–80° (2θ). The specific operations and procedures were described in a previous paper [47].

The surface elements of the pulp were analyzed using XPS (ESCALAB 250Xi, Thermo Fisher Scientific, Waltham, MA, USA). The thermal weight loss curves of the pulp were examined via TGA (STA 449F5, NETZSH, Bavaria, Germany) by increasing the temperature from 30 to 800 °C at 10 °C·min^−1^ under a nitrogen atmosphere [48].

Pulp brightness (TAPPI T 452 om-08), kappa number (TAPPI T 236 om-06), and viscosity (TAPPI T 254 and T 230) were determined using TAPPI methods [49].

## 4. Conclusions

The consumption of phenolic and non-phenolic lignin model species in the reaction with chlorine dioxide was regulated. The preferential consumption of the phenolic lignin model compound was achieved by the differential chlorine dioxide oxidation. The formation of organic chlorides was reduced because of the inhibition of the reaction of chlorine dioxide with the non-phenolic lignin model compound. The reactive depletion pattern of the lignin model species was extended to the pulp fiber. The treated pulp fibers had a higher CrI, higher thermal stability, and a larger O/C ratio compared to the fibers subjected to traditional chlorine dioxide oxidation treatments. Additionally, a more complete lignin signature signal was observed. However, the signals of the phenolic structure were significantly weaker. The pulp obtained via optimized treatment had a higher Kappa number and viscosity at similar pulp brightness values. In particular, AOX production was limited. The results showed that the bleachability and properties of paper pulp were improved by differential chlorine dioxide oxidation. This provides theoretical support for facilitating the clean and efficient chlorine dioxide bleaching of the pulp.

## Figures and Tables

**Figure 1 ijms-23-13310-f001:**
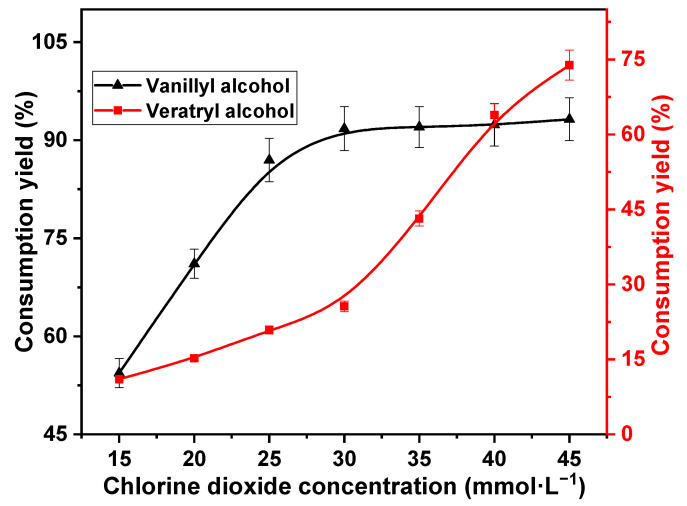
Effect of chlorine dioxide concentration on vanillyl alcohol and veratryl alcohol consumption.

**Figure 2 ijms-23-13310-f002:**
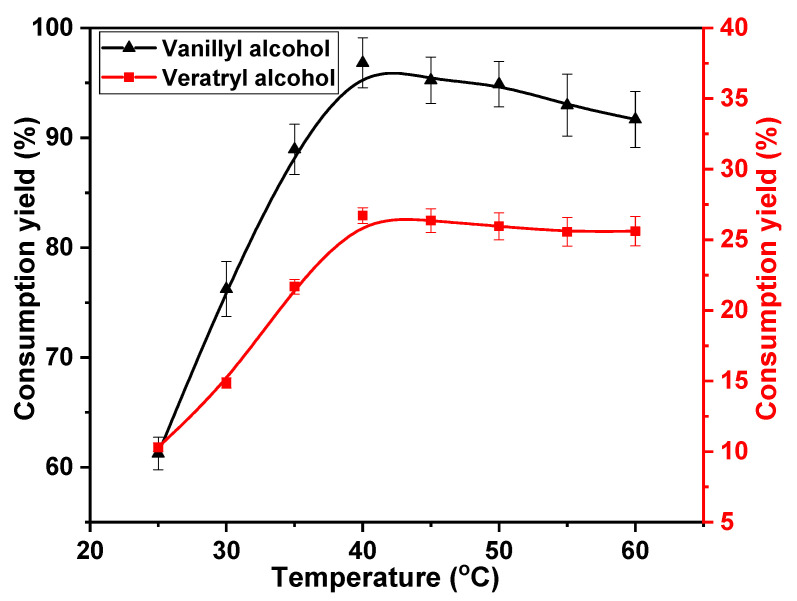
Effect of reaction temperature on vanillyl alcohol and veratryl alcohol consumption.

**Figure 3 ijms-23-13310-f003:**
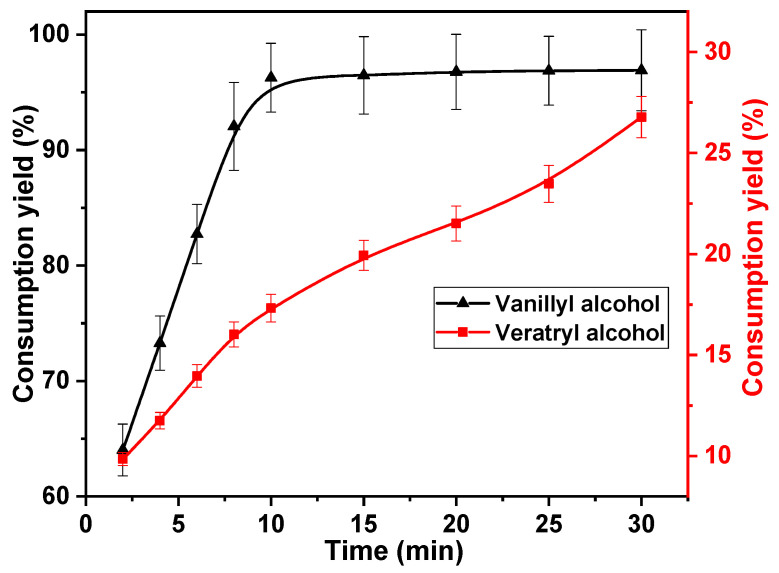
Effect of reaction time on vanillyl alcohol and veratryl alcohol consumption.

**Figure 4 ijms-23-13310-f004:**
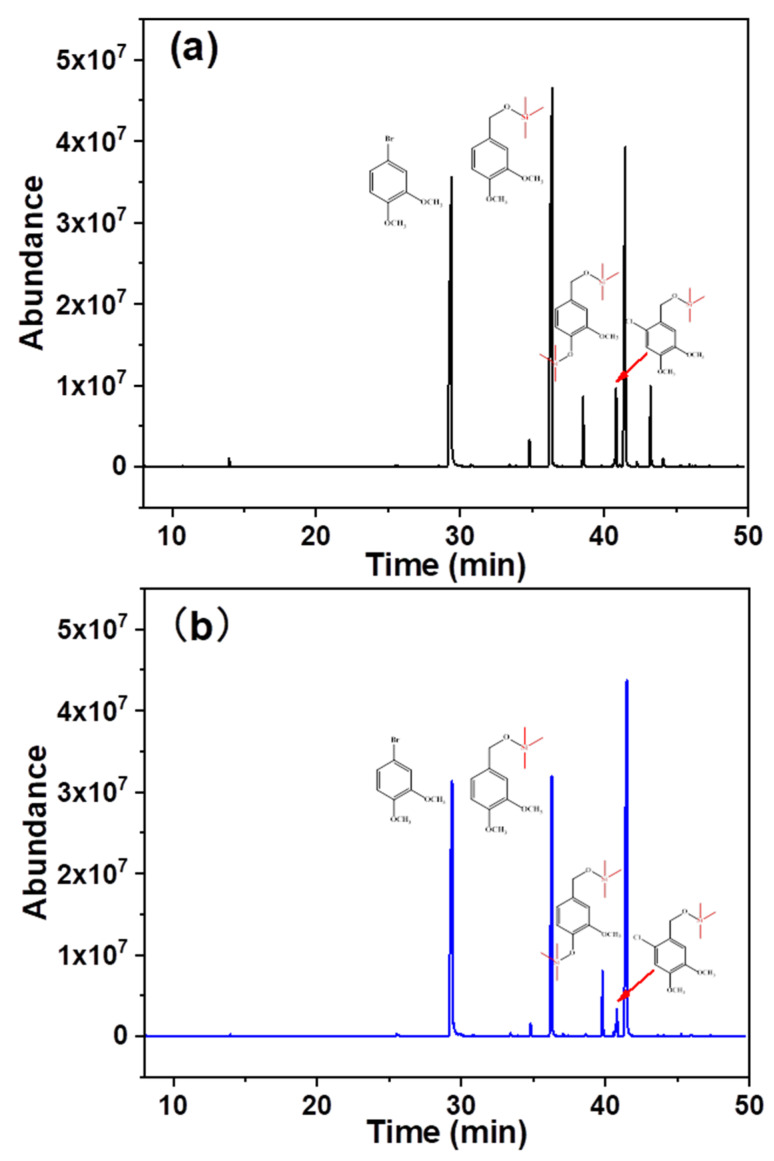
Product composition under different chlorine dioxide oxidation conditions (**a**), traditional chlorine dioxide oxidation; (**b**), optimized oxidation of chlorine dioxide.

**Figure 5 ijms-23-13310-f005:**
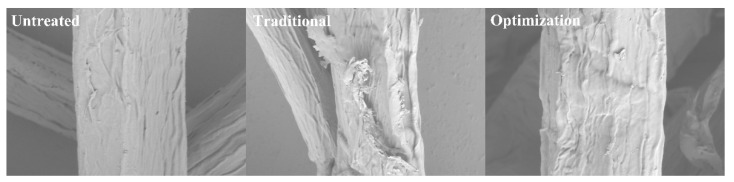
Morphological changes of pulp under different chlorine dioxide oxidation conditions.

**Figure 6 ijms-23-13310-f006:**
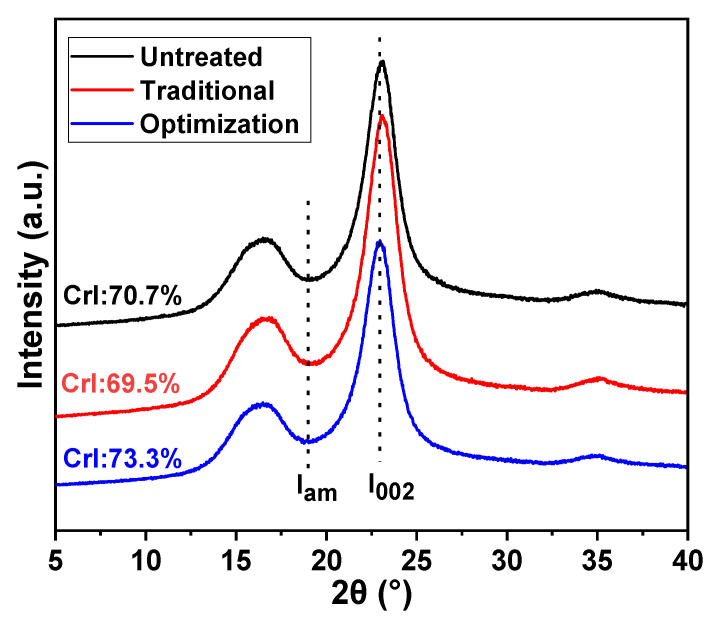
Distribution of crystalline and non-crystalline regions of eucalyptus pulp fiber under different chlorine dioxide oxidation conditions.

**Figure 7 ijms-23-13310-f007:**
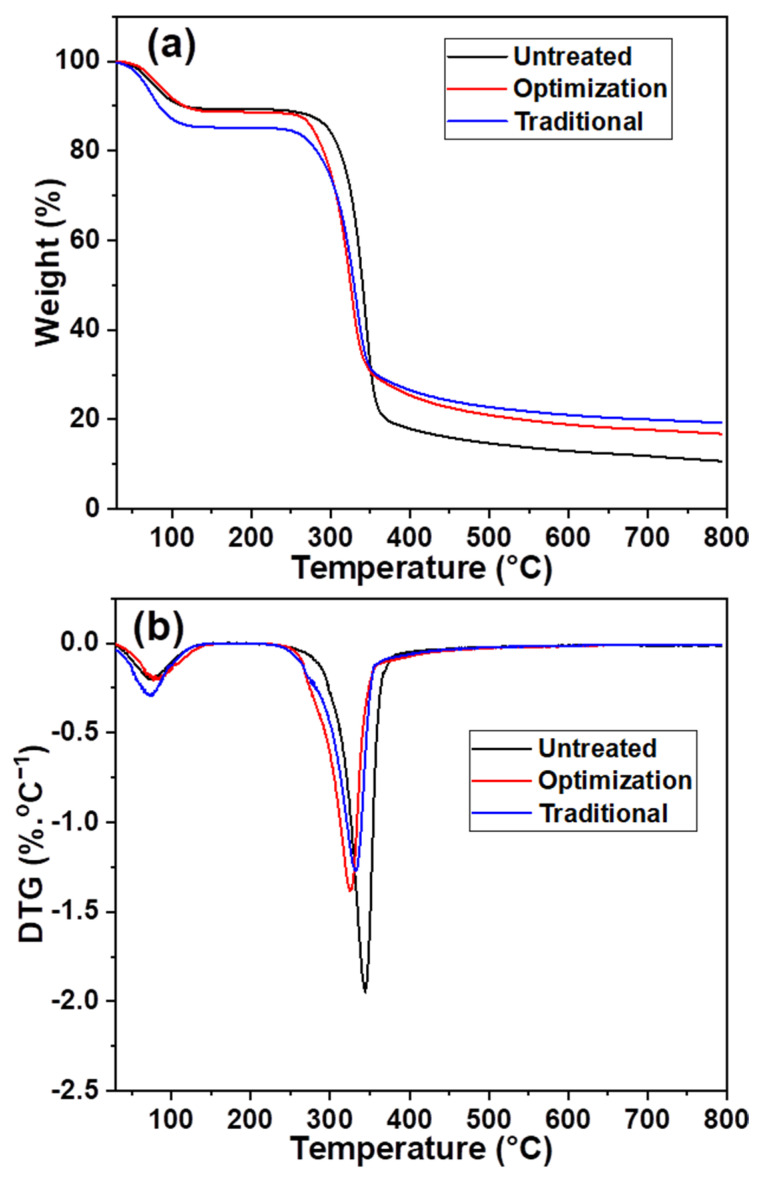
Effects of different chlorine dioxide oxidation conditions on thermal stability of the eucalyptus fiber (**a**), variation of weight loss in different eucalyptus fibers; (**b**), variation of weight loss rate in different eucalyptus fibers.

**Figure 8 ijms-23-13310-f008:**
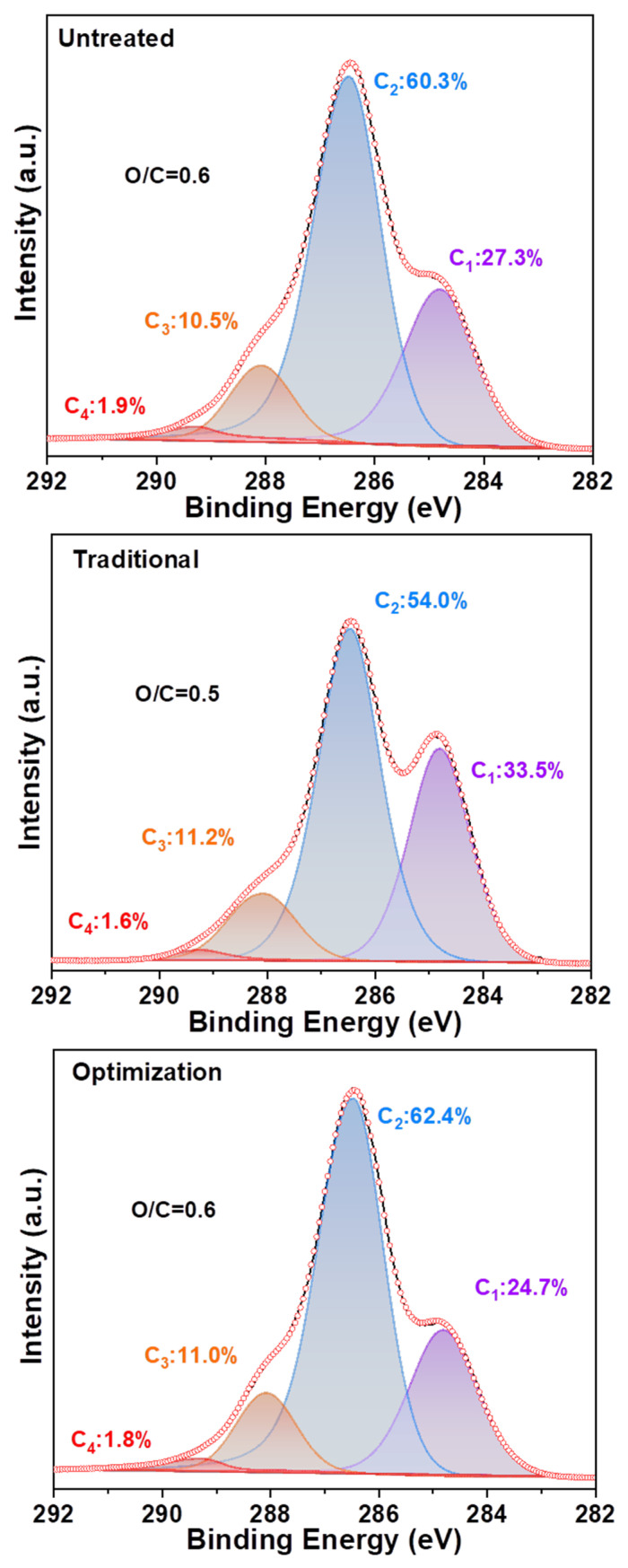
Carbon distribution and oxygen/carbon ratio on the fiber surface of eucalyptus.

**Figure 9 ijms-23-13310-f009:**
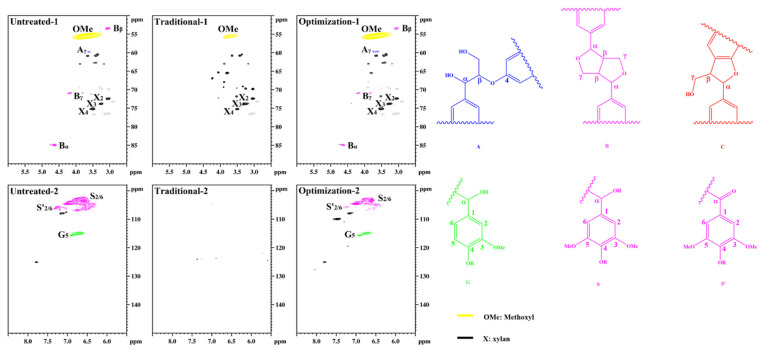
Structural characteristics of lignin in eucalyptus pulp.

**Table 1 ijms-23-13310-t001:** Hydrolysate components under different chlorine dioxide oxidation conditions.

Serial Number	Retention Time (min)	Relative Content		Substance Name
		Traditional Chlorine Dioxide Oxidation	Optimization Chlorine Dioxide Oxidation	
1	29.40	1	1	3,4-dimethoxybromobenzene
2	36.39	0.6	1.0	veratryl alcohol
3	38.53	0.1	0.1	vanillyl alcohol
4	41.46	0.5	0.3	6-chloro-veratryl alcohol

**Table 2 ijms-23-13310-t002:** Different pulp properties and AOX contents in bleaching wastewater.

Samples	Brightness (%ISO)	Kappa Number	Viscosity (mPa·s)	AOX (mg·L^−1^)
Untreated	29.0	19.2	1014	/
Traditional	39.2	7.5	772	23.8
Optimization	41.0	8.4	834	18. 0

## Data Availability

The data presented in this study are available in the manuscript’s figure.

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
