# Peer review of "Effects of the Preferential Oxidation of Phenolic Lignin Using Chlorine Dioxide on Pulp Bleaching Efficiency"

_ijms, 2022, doi:10.3390/ijms232113310_

Round 1

Author Response

Dear Reviewer,

Thank you for your letter and for the comments concerning our manuscript entitled “Effects of the preferential oxidation of phenolic lignin using chlorine dioxide on pulp bleaching efficiency”. We have studied your comments carefully and have made corrections which we hope could meet your requirements. All changes were marked up using the “Track Changes” function.

Questions you put forward are explained as follows:

(1) Line 28: « An efficient and clean bleaching process of the pulp was developed ». I do not agree since AOX are still produced but in a lower amount. Moreover, this paper demonstrates that process conditions (as ClO2 concentration, temperature, …) should be optimised in order to decrease the AOX but this is not the development of a new green process. This should be moderate.

The inappropriate descriptions were modified in accordance with the comments.

The cleaner production processes for chlorine dioxide bleaching were promoted.

(2) Line 94: how were selected the quantity of vanillyl alcohol and veratryl alcohol? In the paper the same quantity for both models has been selected (10 mmole/l) whereas the proportions of phenolic and non-phenolic lignin in Kraft pulp are not equal. The competition of the ClO2 with both models is probably different if on model is in an excess compared to the other one as in the pulp.

In fact, the proportion of phenolic and non-phenolic lignin in unbleached pulp varies significantly with the type of raw material and cooking method. Lucia et al. found that a striking feature of residual lignin in unbleached pulp was the number of phenolic groups, which accounted for about 30-50% of all phenylpropane units of residual lignin. The same quantity of vanillyl alcohol and veratryl alcohol were selected. This is to maximize their competitive reaction characteristics with chlorine dioxide.

Brogdon, BN; Mancosky, DG; Lucia, LA. New insights into lignin modification during chlorine dioxide bleaching sequences (I): Chlorine dioxide delignification. Journal of Wood Chemistry and Technology, 2004, 24(3): 201-219.

(3) Part 2.1: I think that analyses are missing such as pH measurement and residual ClO2 as well as ClO2- and ClO3- after reaction. Indeed, the ClO2 reaction with lignin is complex and pH dependent. The variation of the ClO2 concentration modifies the pH and thus the reactivity with lignin. I think that the complexity of the chlorinated species during lignin (phenolic and non-phenolic) oxidations should be deeper investigated through pH measurements and chlorinated species titration. This was not done in this study.

The pH of the reaction solution was adjusted to 3.5 by adding a pH buffer solution. This is attributed to ineffective decomposition of chlorine dioxide being reduced between pH 3.0 and 4.0. Therefore, the effect of pH change on the reaction of lignin with chlorine dioxide was avoided. In addition, the reaction law of chlorination is considered constant at the same pH. The effect of chlorination on the oxidation of lignin by chlorine dioxide was neglected.

(4) Figure 4: in the title of the figure « a) traditional chlorine dioxide oxidation » - what does it mean? Which conditions were applied for the « traditional conditions »? It is not clearly presented in the material and method section (line 341). Only a reference is given. The authors should precise the conditions.

The details of the chlorine dioxide oxidation reaction were added.

The same amount of the lignin model compounds (10 mmol·L-1 vanillyl alcohol and 10 mmol·L-1 veratryl alcohol) were added to 50 mmol·L-1 chlorine dioxide. The reaction was performed at 70 °C for 60 min.

(5) Table 1: correct in the last column “veratryl alcohol” and “6 chloro- veratryl alcohol”

It was changed as suggested.

(6) Part 2.3 – analysis of physicochemical properties of pulp fibres – I am surprise by the conclusions on morphological changes (SEM) and TGA (but I am not specialised on these analyses): to my knowledge the variation of the ClO2 bleaching conditions as proposed by the authors should not have an impact on the macroscopic level of the fibres, but should impact the chemical composition only (variation in residual lignin quantity, ie kappa number, or on cellulosic chain). Are you sure that the differences in SEM, TGA and crystallinity are significant? I am not convinced.

In principle, chlorine dioxide only affects the chemical component of the pulp (lignin). However, the pH of the reaction solution was adjusted to 3.0 to 4.0 to minimize the ineffective decomposition of chlorine dioxide. This results in acid degradation of hemicellulose and cellulose in pulp. The bleaching loss of pulp was 5.0% during the chlorine dioxide bleaching process. Therefore, the oxidation of chlorine dioxide affects the macroscopic morphology of pulp fibers.

(7) Line 225-226 “the results indicate that the structural integrity of the fibres was enhanced owing to the optimised chlorine oxidation treatment”. I am not convinced by the demonstration which is done before. To my opinion, there is not result to support this conclusion.

The inappropriate descriptions have been changed in revisions.

The results showed that fiber damage was reduced due to selective oxidation of lignin structure using optimized chlorine dioxide oxidation treatment.

(8) Line 234 “This is due to acidolytic degradation of cellulose” – What is the pH and the temperature? To conclude that cellulose is degraded by acidolysis, strong acidic conditions should prevail. Are you sure about that? Generally, it is stated that ClO2 bleaching is selective and that cellulose is poorly affected. Unfortunately, this could not be verified with cellulose viscosity since the authors did not give the cellulose viscosity of the untreated pulp (the value is missing in table 2).

In fact, a small amount of cellulose in the amorphous zone is acidic degraded during the acidic chlorine dioxide oxidation process. The pH and temperature of the chlorine dioxide oxidation process were added. In addition, the viscosity of the unbleached pulp was added. Its decrease indicates the presence of acidic degradation of cellulose.

(9) Line 245 – Again cellulose acid degradation is evoked but I am not convinced by the fact that the conditions are enough severe to lead to acidolysis. I think the comment is not right.

In fact, a small amount of cellulose in the amorphous zone is acidic degraded during the acidic chlorine dioxide oxidation process. The pH and temperature of the chlorine dioxide oxidation process were added. In addition, the viscosity of the unbleached pulp was added. Its decrease indicates the presence of acidic degradation of cellulose.

(10) Table 2 – I think it could be more logical to start the part 2.3 by the table 2 (kappa number, viscosity) and then after to present SEM, crystallinity and TGA. However, NMR analysis can end the part 2.3. Be careful with data: one digit after coma is enough for brightness and kappa number (check accuracy).

It was modified in accordance with the comments.

(11) Line 314- “This had a protective effect on pulp fibres and resulted in an increase in bond strength between fibres, leading to an increase of pulp viscosity”. I do not agree with this sentence. Pulp viscosity is due to the length of cellulose chains, not to the fibre bonding!

The description has been modified.

In particular, ineffective degradation of cellulose was reduced by rapid selective delignification. Its molecular chain was protected. The viscosity of the pulp fiber increases compared with the conventional chlorine dioxide oxidation.

(12) Line 323 – the ClO2 concentration is high. I guess it is an industrial production? Precise it and from which process ClO2 is produced at the mill.

The effective content of chlorine dioxide solution was redetermined. It is produced using the sodium chlorate electrolysis process.

(13) Line 334-339 – the HPL analysis conditions are not enough detailed (eluent, …)

HPL analysis conditions were improved. In particular, the details of elution conditions were added.

Elution conditions was ultrapure water (with 0.1% formic acid) and acetonitrile (90/10, v/v), elution time was 20 min.

(14) Line 240 – “traditional chlorine dioxide conditions” – the conditions should be given; the ref is not enough.

The experimental details of traditional chlorine dioxide oxidation were added.

(15) Line 355-360 – The conditions of traditional and differential chlorine dioxide oxidations are not given: ClO2 charge/ pulp (or chlorine factor?), temperature, time, initial pH. This should be detailed.

The conditions of traditional and differential chlorine dioxide oxidations were added.

The traditional chlorine dioxide oxidation was 20 g pulp (dry basis), pulp/water (1/10, v/v), available chlorine concentration of 2.0%, pH 3.5, 70 °C for 60 min.

The differential chlorine dioxide oxidation was 20 g pulp (dry basis), pulp/ water (1/10, v/v), available chlorine concentration of 0.5%, initial pH 3, 40 °C for 30 min.

(16) Line 362-366 – How lignin was extracted? The method should be presented. Be careful, after such low kappa number (7.5 and 8.4 after ClO2 bleaching), the extraction yield is generally very low because this lignin is difficult to extract, meaning that the extracted lignin does not reflect the residual lignin inside the pulp.

In fact, lignin in pulp before and after chlorine dioxide oxidation was extracted using the milled wood lignin (MWL) method. However, the difficulty of lignin extraction was increased due to the low lignin content in the pulp after chlorine dioxide oxidation treatment. Fortunately, lignin of sufficient content and structural integrity was extracted by MWL treatment of large quantities of pulp.

(17) Line 382 – “the production of organic chlorides was suppressed”. I do not agree, it was “limited”

It was modified in accordance with the comments.

The formation of organic chlorides was reduced because of the inhibition of the reaction of chlorine dioxide with the non-phenolic lignin model compound.

(18) Line 385-387 – I am not convinced by the fact that the variation of the conditions for ClO2 bleaching proposed by the authors has an impact on the macroscopic level of the fibre.

The obvious effect of chlorine dioxide oxidation on the macroscopic level of pulp fibers was confirmed.

In principle, chlorine dioxide only affects the chemical component of the pulp (lignin). However, the pH of the reaction solution was adjusted to 3.0 to 4.0 to minimize the ineffective decomposition of chlorine dioxide. This results in acid degradation of hemicellulose and cellulose in pulp. The bleaching loss of pulp was 5.0% during the chlorine dioxide bleaching process. Therefore, the oxidation of chlorine dioxide affects the macroscopic morphology of pulp fibers.

(19) Line 391 – « The results indicate that the fibre stability is improved » - What does « fibre stability » mean? it is not clear nor supported by data.

The description has been modified.

The results showed that the bleasability and physicochemical properties of the pulp fibers were improved by differential chlorine dioxide oxidation.

(20) Line 390-391 – “AOX production was suppressed” – I do not agree – see table 2, AOX was reduced from 23.75 mg/l to 18.00 mg/l only! AOX were not suppressed!

It was modified as suggested.

In particular, AOX production was limited.

Reviewer 2 Report

Sections of the article are interchanged in the text. First, the methods and materials should be described, then the experiment itself, and then a discussion of the results of the study.
The introductory part is chaotic and does not give a clear idea of ​​the benefits of these studies.
Model systems and the influence of only one reagent are studied, while in practice all factors act simultaneously. This refers to phenolic (vanillyl alcohol) and non-phenolic (veratryl alcohol).
The conclusions of the article are not clear. It is not clear how to remove the phenolic part of lignin without affecting the rest of its components.

Author Response

Dear Reviewer,

Thank you for your letter and for the comments concerning our manuscript entitled “Effects of the preferential oxidation of phenolic lignin using chlorine dioxide on pulp bleaching efficiency”. We have studied your comments carefully and have made corrections which we hope could meet your requirements. All changes were marked up using the “Track Changes” function.

Questions you put forward are explained as follows:

Sections of the article are interchanged in the text. First, the methods and materials should be described, then the experiment itself, and then a discussion of the results of the study. The introductory part is chaotic and does not give a clear idea of the benefits of these studies. Model systems and the influence of only one reagent are studied, while in practice all factors act simultaneously. This refers to phenolic (vanillyl alcohol) and non-phenolic (veratryl alcohol). The conclusions of the article are not clear. It is not clear how to remove the phenolic part of lignin without affecting the rest of its components.

Thank you for your valuable advice.

First of all, the submitted manuscript follows the submission template. It is formatted in the order of introduction, results and discussion, materials and methods, and conclusion.

The significance of the study is refined in the introduction. Therefore, the phenolic structure of lignin is preferentially consumed through the regulation of the reaction environment. The pollution load of bleaching wastewater is reduced when efficient bleaching is achieved. In addition, this is conducive to the improvement of physical and chemical properties of pulp fiber. This study will be of great significance to reduce pulp bleaching cost and realize cleaner production.

The effects of reactant concentration, temperature and time in model system were investigated. The differential oxidation rates of phenolic and non-phenolic lignin with chlorine dioxide were utilized. The rapid oxidative degradation of phenolic lignin was achieved by regulating reaction conditions. In fact, the structural transformation of non-phenolic lignin has not been investigated during rapid phenolic oxidative degradation. It is future research after this study.

Round 2

Reviewer 1 Report

see the attached document

Author Response

Dear Reviewer,

Thank you for your letter and for the comments concerning our manuscript entitled “Effects of the preferential oxidation of phenolic lignin using chlorine dioxide on pulp bleaching efficiency”. We have studied your comments carefully and have made corrections which we hope could meet your requirements. All changes were marked up using the “Track Changes” function.

Questions you put forward are explained as follows:

(1) Line 94: how were selected the quantity of vanillyl alcohol and veratryl alcohol? In the paper the same quantity for both models has been selected (10 mmole/l) whereas the proportions of phenolic and non-phenolic lignin in Kraft pulp are not equal. The competition of the ClO2 with both models is probably different if on model is in an excess compared to the other one as in the pulp.

It was changed as suggested.

In fact, equimolar quantities were investigated to maximize the competition of the two models with chlorine dioxide.

(2) Part 2.1: I think that analyses are missing such as pH measurement and residual ClO2 as well as ClO2- and ClO3- after reaction. Indeed, the ClO2 reaction with lignin is complex and pH dependent. The variation of the ClO2 concentration modifies the pH and thus the reactivity with lignin. I think that the complexity of the chlorinated species during lignin (phenolic and non-phenolic) oxidations should be deeper investigated through pH measurements and chlorinated species titration. This was not done in this study.

It was changed as suggested.

The pH of the reaction solution was adjusted to 3.5. The decomposition of chlorine dioxide is limited.

(3) Table 1: correct in the last column “veratryl alcohol” and “6 chloro- veratryl alcohol”

It was changed as suggested. They were corrected.

(4) Part 2.3 – analysis of physicochemical properties of pulp fibres – I am surprise by the conclusions on morphological changes (SEM) and TGA (but I am not specialised on these analyses): to my knowledge the variation of the ClO2 bleaching conditions as proposed by the authors should not have an impact on the macroscopic level of the fibres, but should impact the chemical composition only (variation in residual lignin quantity, ie kappa number, or on cellulosic chain). Are you sure that the differences in SEM, TGA and crystallinity are significant? I am not convinced.

Related descriptions have been modified according to comments.

The bleaching loss of pulp was 5.0% during the chlorine dioxide bleaching process. The bleaching loss of pulp was 5.0% during the chlorine dioxide bleaching process. Therefore, changes in fiber microstructure were hypothesized.

(5) Line 225-226 “the results indicate that the structural integrity of the fibres was enhanced owing to the optimised chlorine oxidation treatment”. I am not convinced by the demonstration which is done before. To my opinion, there is not result to support this conclusion.

The speculation was removed.

(6) Line 234 “This is due to acidolytic degradation of cellulose” – What is the pH and the temperature? To conclude that cellulose is degraded by acidolysis, strong acidic conditions should prevail. Are you sure about that? Generally, it is stated that ClO2 bleaching is selective and that cellulose is poorly affected. Unfortunately, this could not be verified with cellulose viscosity since the authors did not give the cellulose viscosity of the untreated pulp (the value is missing in table 2).

It was modified in accordance with the comments.

This is due to the depolymerization of cellulose, accompanied by the oxidative removal of lignin under traditional chlorine dioxide oxidation conditions.

(7) Line 245 – Again cellulose acid degradation is evoked but I am not convinced by the fact that the conditions are enough severe to lead to acidolysis. I think the comment is not right.

It was modified in accordance with the comments.

This is due to the depolymerization of cellulose, accompanied by the oxidative removal of lignin under traditional chlorine dioxide oxidation conditions.

(8) Be careful, line 218-219, you wrote “the main application of chlorine dioxide is to enhance the whiteness and viscosity of pulps”. This is not correct; chlorine dioxide does not enhance the viscosity of pulps. You should write “the main application of chlorine dioxide is to enhance the whiteness while preserving the viscosity of pulps. Moreover in pulp science, we rarely use the word “whiteness”, we prefer the word “brightness”.

Line 232, replace “inhibited” by “limited” because you still have AOX but in a lower amount.

Be careful with data: one digit after coma is enough for brightness, kappa number and AOX (check accuracy) in table 2. To be modified.

Line 225 “more non-phenolic lignin was intercalated between pulp fibers”. I do not understand.

It was modified in accordance with the comments.

This was attributed to the high reactivity of phenolic lignin, and more non-phenolic lignin was retained in pulp fibers.

(9) Line 314- “This had a protective effect on pulp fibres and resulted in an increase in bond strength between fibres, leading to an increase of pulp viscosity”. I do not agree with this sentence. Pulp viscosity is due to the length of cellulose chains, not to the fibre bonding!

The description has been modified.

The viscosity of the pulp fiber is less impacted compared with the conventional chlorine dioxide oxidation.

(10) Line 355-360 – The conditions of traditional and differential chlorine dioxide oxidations are not given: ClO2 charge/ pulp (or chlorine factor?), temperature, time, initial pH. This should be detailed.

This is not correctly written. My proposal: “The traditional chlorine dioxide

oxidation was carried out on 20 g pulp (dry basis) at 10% fiber concentration

using 2.0% chlorine dioxide (w ClO2/w pulp), at pH 3.5 and 70 °C for 60 min.

The differential chlorine dioxide oxidation was carried out on 20 g pulp (dry

basis), at 10% fiber concentration using 0.5% chlorine dioxide (w ClO2/w pulp), at pH 3.5 and 40 °C for 30 min.”

It was modified in accordance with the comments.

(11) Line 362-366 – How lignin was extracted? The method should be presented. Be careful, after such low kappa number (7.5 and 8.4 after ClO2 bleaching), the extraction yield is generally very low because this lignin is difficult to extract, meaning that the extracted lignin does not reflect the residual lignin inside the pulp. You can give the yield of lignin which has been extracted, this will be a proof that lignin was efficiently extracted. Moreover, I think you have no proof that the structural integrity was conserved. Anyway, you can make the hypothesis.

It was modified in accordance with the comments.

The extraction yield of lignin was 48.4%. The structural integrity of lignin is hypothesis.

(12) Line 385-387 – I am not convinced by the fact that the variation of the conditions for ClO2 bleaching proposed by the authors has an impact on the macroscopic level of the fibre.

Such speculative conclusions were removed.

(13) Line 391 – « The results indicate that the fibre stability is improved » - What does « fibre stability » mean? it is not clear nor supported by data. I do not see any data or proof of better physicochemical properties. Be careful you wrote “bleasability” instead of “bleachability” I guess….

The description has been modified.

The results showed that the bleachability and properties of paper pulp were improved by differential chlorine dioxide oxidation.

(14) A final comment: be careful to the number of digits in results. For example, in the part concerning the consumption of the model compounds, you wrote 73.86% consumption (line 117). I think you cannot give such precision… 73.9 is largely enough! See the presented values all along the text and correct.

The number of digits in results was corrected.

Reviewer 2 Report

Your work is suitable for publication in a journal. But in my opinion, it is not entirely correct to say that the viscosity of cellulose increases, since it still falls relative to the untreated fiber. it is not explained why there is a difference in the viscosity of cellblose solutions depending on the type of treatment.

Author Response

Dear Reviewer,

Thank you for your letter and for the comments concerning our manuscript entitled “Effects of the preferential oxidation of phenolic lignin using chlorine dioxide on pulp bleaching efficiency”. We have studied your comments carefully and have made corrections which we hope could meet your requirements. All changes were marked up using the “Track Changes” function.

Questions you put forward are explained as follows:

Your work is suitable for publication in a journal. But in my opinion, it is not entirely correct to say that the viscosity of cellulose increases, since it still falls relative to the untreated fiber. It is not explained why there is a difference in the viscosity of cellulose solutions depending on the type of treatment.

Thank you for your valuable advice. Inappropriate descriptions have been modified. In addition, the reason for viscosity change was added.

The viscosity of the pulp fiber is less impacted compared with the conventional chlorine dioxide oxidation.

In principle, chlorine dioxide only affects the chemical component of the pulp (lignin). However, the pH of the reaction solution was adjusted to 3.0 to 4.0 to minimize the ineffective decomposition of chlorine dioxide. This results in the depolymerization of cellulose and hemicellulose in the pulp. The bleaching loss of pulp was 5.0% during the traditional chlorine dioxide bleaching. However, carbohydrate depolymerization is limited during optimizing the chlorine dioxide optimized treatment process. This was attributed to shorter reaction times. As a result, the viscosity of the pulp increases.

Round 3

Reviewer 1 Report

no further comment